# Curcumin and Ethanol Effects in Trembler-J Schwann Cell Culture

**DOI:** 10.3390/biom12040515

**Published:** 2022-03-29

**Authors:** Lucia Vázquez Alberdi, Gonzalo Rosso, Lucía Velóz, Carlos Romeo, Joaquina Farias, María Vittoria Di Tomaso, Miguel Calero, Alejandra Kun

**Affiliations:** 1Laboratorio de Biología Celular del Sistema Nervioso Periférico, Departamento de Proteínas y Ácidos Nucleicos, Instituto de Investigaciones Biológicas Clemente Estable, Montevideo 11600, Uruguay; 2Max-Planck-Institute for the Science of Light, 91058 Erlangen, Germany; gonzalo.rosso@mpl.mpg.de; 3Max-Planck-Zentrum für Physik und Medizin, 91058 Erlangen, Germany; 4Institute of Physiology II, University of Münster, 48149 Münster, Germany; 5Sección Biología Celular, Facultad de Ciencias, Universidad de la República, Montevideo 11400, Uruguay; lveloz@fcien.edu.uy; 6Departamento de Proteínas y Ácidos Nucleicos, Instituto de Investigaciones Biológicas Clemente Estable, Montevideo 11600, Uruguay; carlosj.romeo@gmail.com; 7Espacio de Biología Vegetal del Noreste, Centro Universitario de Tacuarembó, CENUR Noreste, Universidad de la República, Tacuarembó 45000, Uruguay; joaquina.farias@cut.edu.uy; 8Departamento de Genética, Instituto de Investigaciones Biológicas Clemente Estable, Montevideo 11600, Uruguay; marvi@iibce.edu.uy; 9Unidad de Encefalopatías Espongiformes, UFIEC, CIBERNED, Instituto de Salud Carlos III, 28029 Madrid, Spain; mcalero@isciii.es; 10Queen Sofia Foundation Alzheimer Center, CIEN Foundation, 28031 Madrid, Spain; 11Sección Bioquímica, Facultad de Ciencias, Universidad de la República, Montevideo 11400, Uruguay

**Keywords:** CMT1E, Trembler-J, curcumin, ethanol, Hsps, autophagy

## Abstract

Charcot-Marie-Tooth (CMT) syndrome is the most common progressive human motor and sensory peripheral neuropathy. CMT type 1E is a demyelinating neuropathy affecting Schwann cells due to peripheral-myelin-protein-22 (PMP22) mutations, modelized by Trembler-J mice. Curcumin, a natural polyphenol compound obtained from turmeric (*Curcuma longa*), exhibits dose- and time-varying antitumor, antioxidant and neuroprotective properties, however, the neurotherapeutic actions of curcumin remain elusive. Here, we propose curcumin as a possible natural treatment capable of enhancing cellular detoxification mechanisms, resulting in an improvement of the neurodegenerative Trembler-J phenotype. Using a refined method for obtaining enriched Schwann cell cultures, we evaluated the neurotherapeutic action of low dose curcumin treatment on the PMP22 expression, and on the chaperones and autophagy/mammalian target of rapamycin (mTOR) pathways in Trembler-J and wild-type genotypes. In wild-type Schwann cells, the action of curcumin resulted in strong stimulation of the chaperone and macroautophagy pathway, whereas the modulation of ribophagy showed a mild effect. However, despite the promising neuroprotective effects for the treatment of neurological diseases, we demonstrate that the action of curcumin in Trembler-J Schwann cells could be impaired due to the irreversible impact of ethanol used as a common curcumin vehicle necessary for administration. These results contribute to expanding our still limited understanding of PMP22 biology in neurobiology and expose the intrinsic lability of the neurodegenerative Trembler-J genotype. Furthermore, they unravel interesting physiological mechanisms of cellular resilience relevant to the pharmacological treatment of the neurodegenerative Tremble J phenotype with curcumin and ethanol. We conclude that the analysis of the effects of the vehicle itself is an essential and inescapable step to comprehensibly assess the effects and full potential of curcumin treatment for therapeutic purposes.

## 1. Introduction

The group of human hereditary peripheral neuropathies, known as Charcot-Marie-Tooth disease (CMT), has a prevalence of 1/2500 [1]. Within the CMT, demyelinating neuropathies (CMT1) have mutations that alter the structural integrity of myelin [2].

The mutations affecting the *pmp22* gene, play a central pathognomonic role in the CMT disease, representing between 60% and 70% of total myelinopathies [3]. PMP22 is highest expressed in the Schwann cells as a 160-amino-acid myelin glycoprotein, of 22 kDa, with four transmembrane domains, representing approximately 5% of the total compact myelin proteins [4]. However, central expression of PMP22 has also been signaled [5,6,7,8,9].

However, PMP22 is ubiquitously expressed in various tissues and organs in addition to the nervous system. PMP22 has been reported to play a role in adhesion and proliferation regulation [10], and described in epithelial cells, where it localizes with tight junctions and forms complexes with integrins and P2X7 channels [11,12,13]. It has been suggested that the regulation of PMP22 expression may also be increased by the action of steroid hormones [3,14], which has contributed to the exploration of hormonal therapies in the treatment of the CMT1A phenotype [14,15,16].

The expression of human PMP22 has been also reported during the proliferative and secretory phase of the menstrual cycle, and PMP22 has been shown to colocalize with alpha-6-integrins both in vitro and in human tissue samples. Thus, PMP22 appears to be associated in the endometrium with both cell adhesion and endometrial differentiation [17]. Up to 5% of all CMT1s integrate the group of CMT1E, myelinopathies caused by different point mutations in the *pmp22* [18].

The Trembler J (TrJ/+) mouse is an animal model of CMT1E [19,20,21,22], carrying the same spontaneous mutation in *pmp22* as that found in a human family [23]. Under normal conditions, only 20% of PMP22 is inserted into the membrane, with chaperone assistance, at the cost of high energy expenditure (by the synthesis of unused PMP22 and for the maintenance of the proteasome machinery in charge of eliminating the protein surplus) [24,25]. In disease, the percentage of myelin that is inserted is even lower, so it is common to find intracellular PMP22 aggregates in the Schwann cells (SCs) of TrJ/+ mice [26], interfering with the regular protein transport. Thus, the peripheral nerve fibers from the neurodegenerative phenotype in TrJ/+ show altered autophagic-lysosomal pathways, PMP22 cytoplasmic aggregation, increased ribosome and translational activity [21,27,28,29,30,31].

One of the mechanisms underlying cellular stress situations is the response by Heat Shock Proteins (Hsps) [32,33,34,35,36]. Heat Shock Factor 1 (HSF1), the main regulator of the Hsps, is activated by mTOR under stress [34,37] and its inhibition prevents autophagosome formation [38]. Hsp27, recognized for its dual role in normal situations and tumor processes [39,40,41,42], has also been pointed out as a possible target of action for neurodegenerative diseases [43]. In stressful situations, such as the accumulation of intracellular proteins, Hsp27 activates and modulates serine/threonine protein kinase B (PKB/AKT) action, mTOR main activator [44,45,46]. It has also been reported that Hsp70, another member of the Hsps family, assists in the processing of PMP22 aggregates in TrJ/+ through the Golgi apparatus and their release into Rab7-positive vesicles to the lysosome [47]. It has been observed, both in vivo and in vitro, that there is co-localization of Hsps with PMP22 aggregates [21,47]. In addition, beneficial effects of autophagy, promoted by chaperones and preventing the accumulation of misfolded PMP22, have been reported in TrJ/+ [48]. In CMT2, histone deacetylase 6 (HDAC6) has also been signaled as a potential therapeutic target for the amelioration of the neurodegenerative phenotype, reversing motor and sensory deficits induced by Hsp27 activation [49].

A possible cellular and molecular perspective on the therapeutics of these until now incurable hereditary conditions may focus on cellular drainage or detoxification promoted on the autophagic-lysosomal and UPS-chaperone pathways, together with inhibition or reduction of the mTOR pathway. Furthermore, decreased energy availability is a common key player for the modulation of these pathways. For this reason, caloric restriction (CR) at the neuromotor level has been proposed as a valid therapeutic approach for the alleviation of neurodegenerative conditions (including peripheral neuropathies) [50,51,52,53]. Our group demonstrated that dietary CR activates canonical autophagic pathways by decreasing the levels of aggregated PMP22 and increasing ribophagy in TrJ/+ and in wild-type (+/+) nerves (manuscript in preparation).

CR can be emulated, under certain conditions, by effector molecules of the aforementioned pathways [54,55,56,57,58]. Among them, curcumin, a polyphenol extracted from *Curcuma longa* (Linnaeus, Species Plantarum 1:2. 1753), has shown promising results [59,60]. This compound is used as an antitumor, anti-inflammatory, antioxidant, among other beneficial effects. This wide spectrum of curcumin applications depends mainly on the dosage used and the time of application of the treatments. For example, at high concentrations, which in culture range from 25 µM to 160 µM, curcumin is used as a potent anti-tumoral [61,62]. At low concentrations, it decreases reactive oxygen species (ROS) (in myoblast cell cultures 4 µM curcumin and SC from 0.001 to 1 µM curcumin) [63,64], show anti-inflammatory effects at decreasing signal transducer and activator of transcription 3 (STAT3) activation (in human multipotent adipose tissue-derived stem, 10 µM curcumin) [65], and promotes autophagy by inhibiting acetyltransferases (glioblastoma multiforme cell line, 10 µM curcumin) [59,66] and cell regeneration (primary myoblast culture, 1 µM curcumin) [67,68]. Interestingly, in TrJ mice, there is evidence that curcumin treatment can improve the neurodegenerative phenotype [69]. On the other hand, CR activates the autophagy process, prevents the formation, and promotes the elimination of PMP22 aggregates in cultured SCs [70]. However, little is known about the effect of curcumin and ethanol (EtOH) used as curcumin vehicle in the modulation of PMP22 aggregates, and whether this effect could activate effector molecules in SCs ameliorating the neurodegenerative condition of TrJ mice.

In the present work, we set up a new method to obtain enriched cultures in +/+ and TrJ/+ SCs to comprehensibly investigate the effects of curcumin and EtOH. After isolation, we studied the effects of low doses of curcumin on proliferating SCs by evaluating the expression of PMP22 and the modulation of proteins involved in autophagy/mTOR (HDAC6, ribosomes) and the heat shock response (HSF1 and Hsp27). The main results obtained allowed us to visualize specific responses associated with the wild-type genotype, which were different from those observed in the neurodegenerative genotype. The effect of alcohol as a vehicle for solubilization of curcumin in the culture medium was especially scrutinized and the possible significance of its impact is discussed, contrasting them with literature data.

## 2. Materials and Methods

### 2.1. Animals

C57BL wild-type (+/+) and Trembler-J (TrJ/+) mice were obtained from the B6.D2-Pmp22^Tr-J^/J reservoir (JAX stock #002504, Jackson Laboratories, Bar Harbor, ME, USA). The colony was maintained and replicated at the Laboratory of Animal Experimentation of the Clemente Estable Biological Research Institute (IIBCE, MEC). The animal experimentation protocol was approved by the CEUA-IIBCE ethics committee (protocol No.: 002a/10/2020). Mice were housed in a controlled environment (12 h light/dark cycle) and a mean temperature of 21 ± 3 °C with free access to food and water. Mice were weaned at 21 days of age. For this work, 3-month-old male mice (+/+, *n* = 3; TrJ/+, *n* = 3) were used.

### 2.2. Primary Culture Enriched in SCs

Sciatic nerve fibers were dissected as previously described [29]. Briefly, after the cervical dislocation, the mouse was placed in the ventral decubitus position and sprayed with 70% EtOH (Cat# D010-00-03, Dorwil, Argentina). Both sciatic nerves were dissected using surgical scissors. The nerves were then immersed in Dulbecco’s Modified Eagle’s Medium (Cat#: DMEM-HSPTA, Capricorn, Ebsdorfergrund, Germany) supplemented with 10% Bovine Serum (FBS, Cat#: 26140079, Gibco™, Waltham, MA, USA); 5 µg/mL-penicillin/5 µg/mL-streptomycin/10 µg/mL-neomycin (PSN 1X, Cat#: 15640055, Gibco™, Waltham, MA, USA) and 2 mM GlutaMAX™ (GlutaMAX 1X, Cat#: 35050061, Gibco™, Waltham, MA, USA). Immediately, the epineurium was removed to reduce fibroblast contamination, and the fascicles and fibers were slightly “teased” under a stereoscopic microscope.

After that, we proceed as described by Rosso et al., (2017) [71]. For the next 10 days, the fibers were cultured at 37 °C and 5% CO_2_ and evaluated by Nikon Diaphot 300 inverted light microscope (Nikon, Tokyo, Japan) for SC and fibroblast (FB) growth. To complete cell-dissociation, on day 11 the fibers were exposed for 24 h in collagenase medium: DMEM, supplemented with 10% FBS; PSN 1X and GlutaMAX^™^ 1X; collagenase 225 µg/mL; and 5 mM CaCl_2_. After 24 h, culture plates were trypsinized and the cell suspension was collected and centrifuged at 250× *g* for 3 min at room temperature (RT) (Labnet Prism™ R Refrigerated Microcentrifuge, Cat#: Z723878, Sigma-Aldrich, Taufkirchen, Germany). The supernatant was discarded, while the pellet was resuspended in DMEM containing: 10% FBS; PSN 1X; GlutaMAX^™^ 1X; 8 µM forskolin (Cat#: F6886, Sigma-Aldrich, Taufkirchen, Germany) and 20 µg/mL bovine pituitary extract (Cat#: P1167, Sigma-Aldrich, Taufkirchen, Germany). This cell suspension was seeded and maintained for 48 h at 37 °C and 5% CO_2_. Immediately afterward, the cultures were trypsinized and the suspension was collected and centrifuged at 250× *g* for 3 min at RT. The pellet was resuspended and seeded on plates previously treated with Poly-D-lysine. Petri dishes were coated with 2 mL of Poly-D-lysine 50 µg/mL (PDL, Cat#: A3890401, Gibco™, Waltham, MA, USA) for 30 min, followed by two washes with phosphate-buffered saline (PBS, 137 mM NaCl, 2.7 mM KCl, 10 mM Na_2_HPO_4_, 1.8 mM KH_2_PO_4_, pH 7.2–7.6). The cell culture was maintained at 37 °C and 5% CO_2_ until sub-confluent growth (80% of the whole plate surface), with medium changes every two days.

The greater adhesion of FBs to the substrate observed is a property that differentiates them from SCs when both cell types are in culture. Thus, a technique has been developed to separate the two cell types when they are in culture, based on their different adhesion to the substrate. Indeed, the percentage of FBs can be decreased by performing the so-called Cold-Jet procedure [72,73,74]. Briefly, the medium is removed and replaced by ice-cold PBS which is slowly added and rapidly aspirated. The FBs, with increased adhesion, remain attached to the Petri dish while the SCs detach. The suspension, thus enriched in SCs, was centrifuged at 250× *g* for 3 min. The supernatant was discarded, and the precipitate was resuspended in medium with forskolin and bovine pituitary extract in Petri dishes coated with PDL. These cultures were maintained at 37 °C and 5% CO_2_ until experiments were performed, with medium changes every 2 days. SCs used in the experiments were not further than passage 2.

### 2.3. Curcumin

Treatment of primary isolated SCs with curcumin was performed by mixing the compound in the culture medium. The starting point was a 10 mM curcumin stock solution (Cat#: C1386, Sigma-Aldrich Taufkirchen, Germany) diluted in 100% EtOH, as reported in the literature [61,75,76,77]. This stock solution was stored at −20 °C until use. All subsequent dilutions were performed with a culture medium.

### 2.4. Determination of Curcumin Concentration

We based the determination of the optimal concentration of curcumin to be used on previous literature reports [61,63,64,66,67,68,77,78]. As a result of the search, the concentration of curcumin was explored between 0.05 and 0.55 µM.

The viability of the enriched cultures was studied by 3-(4,5-Dimethylthiazol-2-yl)-2,5 Diphenyltetrazolium Bromide (MTT) assay. For this purpose, 1 × 10^5^ cells per well were seeded, in a 96-well plate. 24 h later, the medium was removed and replaced by medium containing different concentrations of curcumin (0.05; 0.15; 0.25; 0.35; 0.45; 0.55 µM). The concentrations were evaluated in triplicate and three independent experiments were carried out. The treatment was conducted for six days, with medium changes every 48 h. On the sixth day of treatment, the medium was removed and replaced by culture medium with the reagent MTT (#cat. M6494, Invitrogen, Eugene, OR, USA), work dilution (WD): 0.5 mg/mL. This pale yellow, water-soluble compound is reduced in the presence of living cells by mitochondrial dehydrogenases, precipitating as formazan (violet-blue crystals, insoluble in water) [79]. The cells were incubated for 2 h at 37 °C and 5% CO_2_. Then, the medium was removed and the cells were lysed with DMSO to release and solubilize the formazan crystals followed by absorbance measurements at 570 nm and 650 nm (background) in the Varioskan^®^ (Varioskan^®^ Flash, Thermo Fisher Scientific, Waltham, MA, USA). In addition, the effect on the EtOH vehicle was tested separately. Two controls were generated: negative control (without EtOH and curcumin) and EtOH control (with EtOH, without curcumin).

The optical density was calculated by subtracting the values obtained from 650 nm minus 570 nm from the average of the values corresponding to the control. The percentages of the controls were calculated taking as reference the average of the values of the negative control; while, for the different concentrations of curcumin tested, the value that corresponded to 100% viability was the average of the values of the control with EtOH in the culture medium.

### 2.5. Curcumin Treatment

After determination of the optimal curcumin concentration, determined in 2.4., SC enriched cultures were maintained until growth reached 50–60% confluence. At this point, the treatment was applied for six days under three conditions: negative control (N.ctrl); EtOH control (Et.crtl), and treatment with 0.25 µM curcumin in EtOH (Et+Cu). All plates were kept at 37 °C and 5% CO_2_ during the course of the treatment and the medium was changed every 48 h.

### 2.6. Immunolabeling in Enriched SC Cultures

This qualitative and quantitative assay was performed on 8-field slides, with 500 cells/well seeded in each field. After 24 h, the corresponding treatment/control was applied. For each independent experiment, four slides were used: the technique control (without primary antibodies); N.ctrl, Et.ctrl, and Et+Cu. Cells were fixed with 2.5% paraformaldehyde in PHEM buffer (25 mM HEPES Cat# H4034, 60 mM pipes Cat# P1851, 10 mM EGTA Cat# 03777, 2 mM MgCl_2_ Cat# M8266, Sigma-Aldrich Taufkirchen, Germany, pH 7.2–7.6): 4 °C, for 20 min followed by permeabilization with 0.1% Triton-X-100 (Cat# T8787, Sigma-Aldrich Taufkirchen, Germany) in PHEM buffer for 30 min. Then, the cells were then incubated with the specific antibodies in incubation buffer (IB: glycine 100 mM Cat# G8790, 0.1% BSA Cat# A9647, Sigma-Aldrich Taufkirchen, Germany, in PHEM), for 24 h at 4 °C.

The specific antibodies used were: anti-PMP22 (Cat# ab61220, RRID: AB_944897, Abcam, Cambridge, UK) WD: 1:100; anti-HSF1 (Cat# SMC-476, RRID: AB_2702279, StressMarq Biosciences, Victoria, BC, Canada) WD: 1:200; anti-Hsp25/27 (Cat# SMC-114, RRID: AB_2120775, StressMarq Biosciences, Victoria, BC, Canada) WD: 1:200; anti-HDAC6 (Cat# ab56926, RRID: AB_941882, Abcam, Cambridge, UK) WD 1:50; and anti-ribosomal proteins, WD 1:200. After 24 h, IB was washed at RT and then the nonspecific binding sites of the secondary antibodies were blocked using IB with 5% normal goat serum (NGS) for 30 min, 37 °C. Incubation with the secondary antibodies, for 45 min at RT and in the dark, was preceded by washes in IB at RT. The secondary antibodies and probe used were: Goat Anti-Mouse IgG (H+L) Highly Cross-adsorbed Antibody, Alexa Fluor 488 Conjugated (Cat# A-11029, RRID: AB_138404, Molecular Probes, Eugene, OR, USA) WD: 1:1000, Goat Anti-Rat IgG (H+L) Antibody, Alexa Fluor 555 Conjugated (Cat# A-21434, RRID: AB_141733, Molecular Probes, Eugene, OR, USA) WD: 1:1000; Goat anti-Rabbit IgG (H+L) Cross-Adsorbed Secondary Antibody, Cyanine5 (Cat# A10523, RRID: AB_2534032, Thermo Fisher Scientific, Waltham, MA, USA) WD: 1:1000 y DAPI (4′,6-Diamidino-2-Phenylindole, Dihydrochloride) (Cat# D1306, RRID: AB_2629482, Thermo Fisher Scientific, Waltham, MA, USA) WD: 1:1000. As a negative control for immunolabeling, the culture was incubated with secondary antibodies only (Appendix A).

Washes were then performed with IB and subsequently with PHEM buffer. After that, the coverslips were removed and mounted with Prolong™ Diamond Antifade (Cat#: P36970, Invitrogen, Eugene, OR, USA).

### 2.7. Confocal Microscopy

Cell culture growth and morphology of cell types were performed on a Nikon Diaphot 300 inverted light microscope (Nikon, Tokyo, Japan).

Immunostaining was visualized with a Zeiss LSM 800 confocal microscope (Carl-Zeiss-Strasse, Oberkochen, Germany). At the beginning of the confocal session, the maximum laser photomultiplier levels and voltage were set with the negative controls for each sample, until a few bright non-specific signals started to appear. Then, all images containing specific antibodies were taken under the same conditions, in the same section.

### 2.8. Fluorescence Image Analysis

The images obtained were analyzed with ImageJ (version 1.53b, RRID: SCR_003070) [80]. Quantification of the intensity of the differently labeled proteins was performed by discriminating the nuclear compartment from the cytoplasmic compartment. For the nucleus, binary plane-to-plane masks were established, and based on these, intensity values per unit area were obtained. To obtain the cytoplasm values, the total cell, and nucleus masks were subtracted, resulting in the corresponding cytoplasm mask for each cell. For each of the conditions, 100 cells per genotype were evaluated.

### 2.9. Statistical Analysis

The normality of the data obtained was evaluated by the Shapiro–Wilk test. The comparison between controls and the effects of the same concentration of curcumin between genotypes by MTT was performed using the nonparametric Mann–Whitney U signed-rank test. In addition, for the study of the different curcumin concentrations tested, the non-parametric Kruskal–Wallis test was applied, together with the Dunn’s multiple comparison test, evaluating the differences concerning to Et.crtl.

For the comparison between N.ctrl, Et.ctrl, and Et+Cu in each genotype, one-way ANOVA (F_DFn = 2, DFd = 297_, *p*-value) or the Kruskal–Wallis (H, *p*-value) test were performed. Multiple comparisons were performed to understand the variations between N.ctrl vs. Et.ctrl and Et.ctrl vs. Et+Cu treatment. Dunnett’s test for one-way ANOVA (*p*-value, (95 CI of Dif)) and Dunn’s test for Kruskal–Wallis (*p*-value). In addition, the comparison of each treatment between genotypes was analyzed by unpaired Student’s *t*-test (*p*-value, *t*, df = 198) or the Mann–Whitney U signed-rank test (*p*-value, U). All tests were applied using a two-tailed distribution and the results were considered significant at an alpha level of 0.05. Statistical analysis was performed with GraphPad Prism version 8.0.0 (RRID: SCR_002798, GraphPad Software, San Diego, CA, USA).

## 3. Results

### 3.1. Morphological Differences between FBs and SCs

The identification of both cell types was based on their morphological differences, distinguishable under the bright field microscope. SCs have a fusiform bipolar shape, with a central zone from which a prolongation extends to each end (Figure 1, SC). The extensions are of variable length, but common to all SCs is the presence of one extension longer than the other. SCs are easily distinguishable from FBs as the latter have a heterogeneous, mainly stellate shape (Figure 1, FB).

The spreading area as well as the major and minor axis length values of both cell types were measured and shown in Table 1. Comparison between total spreading areas reveal that one FB possesses an area equivalent to almost 30 times bigger than a SC. These morphological and size differences were visualized in +/+ and TrJ/+ cultures (Figure 1A) and immunolabeled confocal images (Figure 1B). Morphological comparisons of different cell types, between +/+ and TrJ/genotypes, yielded no significant differences (data not shown).

### 3.2. Obtaining of Cultures Enriched in +/+ and TrJ/+ SCs

The mixed cultures obtained from the application of two different protocols are visualized in Figure 2. The “collagenase-trypsin” method allowed to obtaining a large cell volume after 10 days, although it yields a low SC/FB ratio (Figure 2, left panels). On the other hand, the modified method described in Rosso et al., 2017 [55] showed a better SC/FB ratio, although it takes longer (21 days) (Figure 2, right panels).

Although it presents greater complexity, the second protocol contains two key steps where its efficiency lies in yielding a greater number of SCs: namely the initial step which requires an exhaustive removal of the epineurium, and the last stage of the protocol where enrichment of the SCs takes place after using the Cold-Jet technique. Taking into account these advantages, this last protocol was selected over the “collagenase-trypsin” method to continue with the experiments.

### 3.3. Determination of Suitable Curcumin Concentration to Be Used in Enriched +/+ and TrJ/+ SC Cultures

The determination of the lowest curcumin concentration, with no effect on the SC culture viability (i.e., no difference from the Et.ctrl) was evaluated by MTT assay (Figure 3).

The effect of EtOH compared to untreated cultures (Figure 3A), showed in +/+ SCs an increase in viability by 42% (*p* = 0.0275, U = 218), while in SCs TrJ/+ an increase of 33% (*p* = 0.0002, U = 163). In addition, the comparison between +/+ and TrJ/+ of EtOH effects, revealed the absence of significant differences, suggesting that the EtOH effect is similar in both genotypes.

The comparison between Et+Cu concentrations vs. Et.ctrl (Figure 3B) showed a decrease in +/+ SCs viability at 0.05, 0.15, and 0.55 µM concentrations (H = 61.76, *p* < 0.0001; 0.05 µM: *p* = 0.0003; 0.15 µM: *p* = 0.0015; 0.55 µM: *p* = 0.0001); while at 0.25, 0.35, and 0.45 µM we did not observe differences from the Et.ctrl. The results in TrJ/+ SCs showed a decrease in viability at 0.05, 0.15, 0.35 and 0.55 µM concentrations (H = 23.03, *p* = 0.0008; 0.05 µM: *p* = 0.0029; 0.15 µM: *p* = 0.0036; 0.35 µM: *p* = 0.0097; 0.055 µM: *p* = 0.0001), while 0.25 and 0.45 µM had no difference with respect to Et.ctrl.

In summary, these results indicate that 0.25 µM curcumin was the lowest concentration showing no significant differences from the vehicle control in both genotypes and also had no significant differences between genotypes (*p* = 0.0557, U = 319) (Figure 3C).

### 3.4. Effect of Curcumin Treatment on SC +/+ and TrJ/+ Enriched Cultures

#### 3.4.1. Alterations in PMP22 Expression

The study of PMP22 expression in +/+ and TrJ/+ SCs was analyzed by confocal microscopy (Figure 4). At the nuclear level, for both +/+ and TrJ/+ SCs, one-way ANOVA analysis reported significant differences between the N.ctrl, Et.ctrl and Et+Cu treatments (+/+: F = 16.14, *p* = 0.0062); TrJ/+: F = 2.425, *p* = 0.0004). In +/+ SCs, there is a decrease of PMP22 expression in 0.43 mM Et.ctrl, compared to N.ctrl (*p* = 0.0014, (−21.12; −3.875)) and an increase in the 0.25 µM curcumin, compared to Et.ctrl (*p* = 0.0292, (−17.07; −0.8420)). For TrJ/+ SCs, there was only a reported decrease in PMP22 expression in Et.ctrl, compared to N.ctrl (*p* = 0.0292, (−53.08; −12.10)). At the cytoplasmic level, significant differences were reported between N.ctrl, Et.ctrl and Et+Cu treatments (+/+: H = 31.57, *p* < 0.0001); TrJ/+: F = 4.864, *p* < 0.0001) for both +/+ and TrJ/+ SC. In +/+ SCs, a decrease in PMP22 expression with Et.ctrl, compared to N.ctrl (*p* = 0.0343), and an increase in Et+Cu treatment, compared to Et.ctrl (*p* < 0.0001); while in TrJ/+ SC a significant decrease in expression was observed with Et.ctrl, compared to the N.ctrl (*p* = 0.006, (−77.22; −23.58)).

Variations in PMP22 expression levels were also evaluated in different conditions comparing SCs of +/+ and TrJ/+. In nuclear and cytoplasmic domains, N.ctrl showed higher PMP22 expression in TrJ/+ (nucleus: *p* = 0.0097, t = 2.910; cytoplasm: *p* = 0.0020, t = 3.519), while in the Et+Cu treatment, presented higher PMP22 expression in +/+ SC (nucleus: *p* = 0.0096, t = 2.964; cytoplasm: *p* < 0.0001, t = 6.549).

The data indicate an increase in PMP22 expression with curcumin treatment in +/+ SC, compared to Et.ctrl in nucleus and cytoplasm, but no changes are visualized in TrJ/+ SCs. In addition, N.ctrl showed increased expression in TrJ/+, compared to +/+, whereas Et+Cu treatment showed increased PMP22 expression in +/+ SC, in both compartments.

#### 3.4.2. Effect of Curcumin on Heat Shock Response: Expression of HSF1 and Hsp27

The study of HSF1 and Hsp27 expression, as markers of stress response, in +/+ and TrJ/+ SC enriched cultures, modulated by the curcumin treatment, was analyzed by confocal microscopy (Figure 5). The nuclear HSF1 in Figure 5A show significant differences between N.ctrl, Et.ctrl and Et+Cu (+/+: H = 7.046, *p* = 0.0295; TrJ/+: F = 0.28, *p* = 0.0072). In +/+ SCs an increase of HSF1 intensity was observed after Et+Cu treatment, compared to Et.ctrl (*p* = 0.0210). However, we observed a decrease of HSF1 with Et.ctrl compared to N.ctrl (*p* = 0.0205, (−46.82; −3.976)) in TrJ/+ SCs. No difference was observed with respect to Et+Cu treatment. In cytoplasmic domains, HSF1 showed significant differences between N.ctrl., Et.ctrl and Et+Cu (+/+: F = 19.44, *p* < 0.0001; TrJ/+: F = 3.75, *p* = 0.0004); in +/+ SC a decrease of HSF1 with Et+Cu treatment compared to Et.ctrl was evident (*p* < 0.0001, (38.84; 45.36)).

Meanwhile, in TrJ/+ SC a decrease of HSF1 with Et.ctrl, compared to N.ctrl was observed (*p* = 0.0068, ([−55.08; −9.316)). Differences between conditions in both genotypes revealed significant differences only at the cytoplasmic level. Both N.ctrl and Et+Cu treatment showed higher expression of HSF1 in TrJ/+ (N.ctrl: *p* = 0.0374, t = 2.247; Et+Cu treatment: *p* = 0.0134, U = 6.50); while Et.ctrl showed higher expression in +/+ (*p* = 0.0004, t = 4.719).

On the other hand, Hsp27 showed significant differences in the three studied conditions only at the cytoplasmic level (+/+: F = 15.88, *p* = 0.0049; TrJ/+: H = 11.79, *p* = 0.0002), but differences were only observed in +/+ SC, giving an increase in Hsp27 expression with curcumin treatment compared to the Et.ctrl (*p* = 0.0101, (−41.25; −5.288)). Furthermore, differences between +/+ and TrJ/+ in the different conditions are observed only at the cytoplasmic level: while Hsp27 expression of the N.ctrl was higher in TrJ/+ SC (*p* = 0.0230, t = 2.485), curcumin treatment resulted in higher +/+ SC (*p* = 0.0009, t = 2.950) (Figure 5B).

The data indicate a nuclear increase and a cytoplasmic decrease in HSF1 expression, along with a cytoplasmic increase in Hsp27, with curcumin treatment in +/+ SC, compared to the E.ctrl, but no changes are visualized in TrJ/+ SC. Furthermore, at the cytoplasmic level, increased expression of HSF1 and Hsp27 in TrJ/+ SC was observed in N.ctrl, increased expression of HSF1 in +/+ SC in Et.ctrl and increased expression of HSF1 in TrJ/+ and Hsp27 in +/+ SC in Et+Cu.

#### 3.4.3. Effect of Curcumin on Autophagy/mTOR: Expression of HDAC6 and Ribosomes

The expression of HDAC6 and ribosomes, as autophagy/mTOR pathway markers, in SC +/+ and TrJ/+ cultures in response to curcumin treatment was analyzed by confocal microscopy (Figure 6).

In all compartments, HDAC6 showed significant differences between N.ctrl, Et.ctrl and Et+Cu (nucleus: +/+: F = 3.676, *p* = 0.0002; TrJ/+: H = 6.79, *p* = 0.0243; cytoplasm: +/+: F = 2.92, *p* = 0.0006; TrJ/+: F = 1.57, *p* = 0.0066). Moreover, in both nuclear and cytoplasmic compartments the same behavior occurred: in +/+ SC an increase of HDAC6 imprinting was observed with Et+Cu treatment, with respect to the Et.ctrl (nucleus: *p* = 0.0001, (−82.79; −27.14); cytoplasm: *p* = 0.0431 (−29.42; −0.418)); while in TrJ/+ SC a decrease in HDAC6 expression was observed in the Et.ctrl, respect to N.ctrl (nucleus: *p* = 0.0235; cytoplasm: *p* = 0.0138 (−38.95; −4.713)). In nuclear domains, higher HDAC6 expression was observed only in TrJ/+ SC with Et+Cu treatment (*p* = 0.0140, t = 3.655), and at the cytoplasmic level differences in expression were seen with the N.ctrl (*p* = 0.0013, t = 4.5854) and with Et+Cu treatment (*p* = 0.0165, t = 2.597), being in both cases higher expression in +/+ SC (Figure 6A).

Ribosome expression levels showed differences in all compartments between N.ctrl, Et.ctrl and Et+Cu (nucleus: +/+: F = 12.75, *p* = 0.0190; TrJ/+: F = 1.482, *p* < 0.0001; cytoplasm: +/+: H = 21.02, *p* < 0.0001; TrJ/+: F = 4.01, *p* = 0.0057). In the nucleus, the ribosome expression showed differences in +/+ SC, there being an increase in expression with Et+Cu treatment, with respect to the Et.ctrl (*p* = 0.0323 (−8.847; −0.353)), while in TrJ/+ SC the difference translated into a decrease in ribosomal expression in the Et.ctrl, compared to N.ctrl (*p* < 0.0001, (−59.42; −37.25)). In SC +/+ cytoplasm, a decrease in ribosome expression was observed with Et.ctrl, compared to N.ctrl (*p* = 0.0001), and an increase with Et+Cu, compared to Et.ctrl (*p* = 0.0003). In the cytoplasm of TrJ/+ SC only a significant decrease was observed with Et.ctrl (*p* = 0.0055, (−26.59; −5.08)), as was +/+ SC. The evaluation of differential expression within the same condition, for the +/+ and TrJ/+ genotypes, showed, at the SC nuclear level, a higher basal culture conditions’ expression of ribosomes in TrJ/+ (*p* < 0.0001, t = 8.610) and a higher expression in +/+ SC Et.ctrl (*p* = 0.0329, t = 2.335). At the cytoplasmic level, the highest ribosomal expression was observed in +/+ SC with both Et.ctrl (*p* < 0.0001, U = 0) and Et+Cu treatment (*p* = 0.0002, U = 0) (Figure 6B).

The data indicate a nuclear and cytoplasmic increase in HDAC6 and ribosome expression with curcumin treatment in +/+ SC compared to the Et.ctrll; but no changes are visualized in TrJ/+ SC. In addition, for HDAC6 a cytoplasmic increase was observed in N.crtl in TrJ/+ SC and an increase in both compartments in +/+ SC with Et+Cu treatment. In the case of ribosomes, N.crtl indicated increased nuclear expression in TrJ/+ SC, E.crtl increased expression in +/+ SC in both compartments, and Et+Cu increased cytoplasmic expression in +/+ SC.

## 4. Discussion

PMP22 expression has been studied both in vivo, using different animal models [19,25,48,81,82,83,84], and in vitro in SC cultures [24,63,85], respectively. These approaches denote different but complementary physiological conditions. While in vivo approaches allow understanding how and where the main expression of this protein is located in SCs arrested in G0, the in vitro studies allow the evaluation of the expression in those cells that are in a proliferative state. One of the contributions of our work lies in the evaluation of the basal culture conditions for the expression of PMP22 in TrJ/+ SCs, compared with that of +/+, discriminating the nuclear and cytoplasmic compartments. Overall, in both cellular domains, PMP22 expression was higher in TrJ/+ SCs compared to +/+ SCs. This result is in line, not only with works reporting the existence of cytoplasmic aggregates of PMP22 in TrJ/+ nerves [29,69,83,84] but also in agreement with previous work from our group, which determines the PMP22 expression in +/+ and TrJ/+ SCs inside the nucleus [8,9].

The low dose of curcumin treatment was applied as a possible strategy to stimulate cellular detoxification pathways, and thus alleviate the neurodegenerative phenotype. Our curcumin treatment had an additional effect caused by the EtOH vehicle (used to let cur-cumin solubilization for ulterior cell entry), in both +/+ and TrJ/+ SCs. This collateral effect, observed after six days of treatment, led us to inquire about the use and, more importantly, about the validation of the vehicle so that it could be used without specific effects. In this sense, the recommended vehicles for solubilization of curcumin by the manufacturer are ethanol and dimethyl sulfoxide (DMSO). Although in many papers the effects of the sol-vents on culture viability are not shown [52,64,65,86] (or are not discussed [62,87,88]), if they do not present cytotoxic effects, the reality is that both vehicles produce effects that vary widely depending on the cell type and duration of treatment [87,89,90,91]. The pre-established idea of the absence of the effects of these vehicles is recurrent in the literature. However, the contribution of the vehicle prevents us from clearly discriminating the real effect and pharmacological potential of curcumin. For this reason, it must be settled as this constitutes a central point, to circumscribe the results only to the applied treatment. We have tested the impact of DMSO as a vehicle of curcumin on fibroblast from +/+ sciatic nerves and our preliminary results seem to indicate an equivalent effect com-pared to that of EtOH. In our primary results, different concentrations of DMSO were test-ed and in all cases, after five days of DMSO exposure, the viability and proliferation were significant differences compared with the untreated control (Appendix A). In future work, we will seek to determine other strategies that allow solubilizing and targeting curcumin, evaluating at each step the cell viability to corroborate a negligible effect of the vehicle.

The impact of curcumin on the studied pathways in +/+ SC could not be analyzed in the TrJ/+ genotype, because all markers showed no significant difference between EtOH control and curcumin treatment. The results highlight the lability of the TrJ/+ genotype, expressed in its poor capacity to recover from the impact of the vehicle. In the literature, ethanol has been reported to increase ROS and mitochondrial dysfunction [92,93,94]. In zebrafish, at concentrations of 1% ethanol [92], a differential effect in mitochondrial function, with acute and chronic treatment, has been described. However, different performances showed that the mechanisms triggered are also dependent on administration protocols. Since the MTT assay is based on the conversion of tetrazolium to formazan by mitochondrial dehydrogenases, we do not rule out the possibility that ethanol also affects the mitochondria in Schwann cells. Therefore, the increase in the ethanol control relative to the negative control could be due to dehydrogenases’ functionality changes in response to the vehicle, rather than the normal increase in viability. In this sense, the fact that both genotypes showed equivalent responses in ethanol control to the negative control, supports this hypothesis.

In addition, the results found by our group show mitochondrial differences in the nerves of +/+ and TrJ/+ mice. From the analysis of electron microscopy images, we obtained the number of mitochondria per fiber in the axonal and SC domains (Appendix A). These results show that there are differences in the number of mitochondria when comparing SCs of +/+ vs. TrJ/+ fibers. The morphological analysis considering the largest and smallest diameters of mitochondria, also showed apparent differences between +/+ and TrJ/+ (Appendix AA). Although there is a correlation between both parameters for the two genotypes, the equations representing the linear correlation are different (Appendix AB). Furthermore, when looking at the genome expression, a qPCR analysis of the cytochrome b gene transcript level shows a higher amount of the transcript in +/+ compared to TrJ/+ (Appendix AC).

Thus, the study of the vehicle takes on particular relevance for in vitro approaches to the autophagy/mTOR and chaperone pathways in the TrJ/+ neurodegenerative genotype.

In the present work, the effect of ethanol was visualized equally between +/+ and TrJ/+ SCs, and the percentage of viability was calculated taking the ethanol control as 100%. The latter allowed us to obtain the effect of curcumin to analyze the data. Despite the side effect of ethanol, we were able to determine a 0.25 µM curcumin as the lowest concentration that showed no difference in viability to the ethanol control.

From the study of the heat-stress markers’ response pathway, we were able to establish in +/+ SC the HSF1 and Hsp27 expression concordant with that reported in the literature. The HSF1 functions as a transcription factor, which under stress situations is activated and translocated to the nucleus, inducing the expression of the pathway’s effectors such as Hsp27 [95,96,97,98] Furthermore, curcumin treatment allowed us to observe an increase in HSF1 at the nuclear level and a decrease at the cytoplasmic level, together with an increase in Hsp27, indicating a possible activation of this pathway. In addition, HDAC6 expression increased after treatment with curcumin. This increase of the protein, a member of basal autophagy in-volved in the selective elimination of aberrant protein aggregates [99,100] suggests that in the wild-type genotype, curcumin treatment may be stimulating this degradation pathway. Conversely, the increase in ribosomal expression under mTOR regulation [101,102] supports the idea of a favorable nutritional and energetic context in +/+ SC after curcumin treatment.

The impact of curcumin on the studied pathways in +/+ SC could not be analyzed in the TrJ/+ genotype because all of the markers showed no significant difference between the EtOH control and curcumin treatment. These results indicate a TrJ/+ genotype lability that prevents SCs from recovering from the ethanol shock. Thus, the study of the vehicle takes on particular relevance for in vitro approaches to the autophagy/mTOR and chaperone pathways in the TrJ/+ neurodegenerative genotype.

On the horizon of the TrJ/+ in vitro approaches, and its response to the action of different neuroprotective and anti-inflammatory agents, such as curcumin, the analysis of the effects of the vehicle itself is an essential, inescapable, and conditional step for the fine-tuning of the experimental strategy to be applied.

## 5. Conclusions

Our work established a new experimental strategy for obtaining enriched cultures of SC from +/+ and TrJ/+ mice. We were able to determine a curcumin concentration with no effect on viability in both genotypes for an extended period of time, which allowed us to study the expression of key autophagic-pathway markers in the accumulation of PMP22 protein in SCs. We found an intrinsic ethanol effect in +/+ and TrJ/+ SC that was reversed by curcumin treatment in +/+, but not in TrJ/+ SC. These in vitro cultures allow pre-clinical investigations of promising therapeutic strategies or pharmacological compounds such as curcumin, for the alleviation of human-related peripheral neuropathies.

## Figures and Tables

**Figure 1 biomolecules-12-00515-f001:**
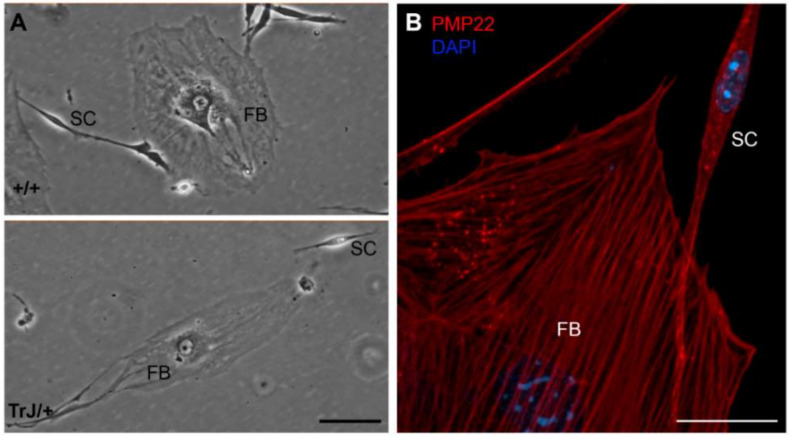
Morphology of different cell types in enriched SC cultures. (**A**) In both +/+ and TrJ/+ genotypes, fibroblasts (FBs) and Schwann cells (SC) show morphological differences. FBs show a heterogeneous flattened shape with abundant cytoplasm and a large nucleus easily distinguishable under light microscopy. SCs show typically bipolar shape with lateral extensions and a smaller size compared to FBs; (**B**) Confocal microscopy image showing PMP22 (red) and nuclear labeling counterstain (DAPI, blue). Scale in (**A**) = 250 µm. Scale in (**B**) = 20 µm.

**Figure 2 biomolecules-12-00515-f002:**
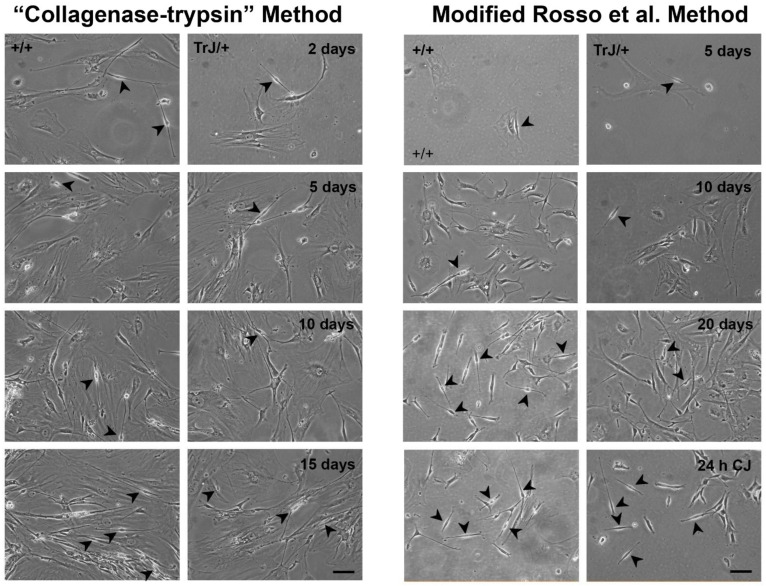
Enriched SC–FB cultures obtained by different protocols. Images of mixed primary culture obtained from the “collagenase-trypsin” protocol are visualized in the left panels in the +/+ and TrJ/+ cultures. This protocol allowed the obtaining of a large number of cells in eight days, but the SC/FB ratio is rather low, as observed after 15 days of seeding. Images of the mixed primary culture obtained with the modified protocol of Rosso et al., 2017 [71] in +/+ and TrJ/+ are displayed in the right panels. This method requires more time to obtain a large number of cell volumes. The epineurium removal and enrichment by Cold-Jet (CJ) allowed us to obtain a higher SC/FB ratio, compared to the collagenase-trypsin protocol. Arrowheads point to SCs. Scale = 250 µm for all panels.

**Figure 3 biomolecules-12-00515-f003:**
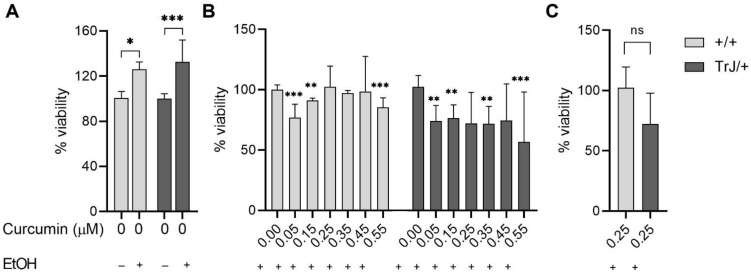
Cell viability by MTT assay. Enriched SCs cultures +/+ and TrJ/+ were treated for six days with different concentrations of curcumin (added to the culture medium) in addition to the evaluation of N.ctrl (without curcumin and EtOH) and Et.ctrl (without curcumin, with EtOH, 0.43 mM). (**A**) Control comparison within each genotype show a significant increase in viability of the Et.ctrl compared to N.ctrl. Comparison +/+ vs. TrJ/+ no significant differences (data not shown); (**B**) Comparison of the different curcumin concentrations to Et.ctrl in +/+ and TrJ/+. (**C**) Comparison +/+ vs. TrJ/+ of the lowest concentration (0.25µM) without differences with respect to Et.ctrl. obtained in (**B**) for both genotypes. The data, expressed as median ± SIR, were analyzed in (**A**,**C**) by Mann–Whitney U signed-rank test and in (**B**) by the Kruskal–Wallis test. * *p* < 0.05 ** *p* < 0.002; *** *p* < 0.0002. SIR: semi-interquartile range.

**Figure 4 biomolecules-12-00515-f004:**
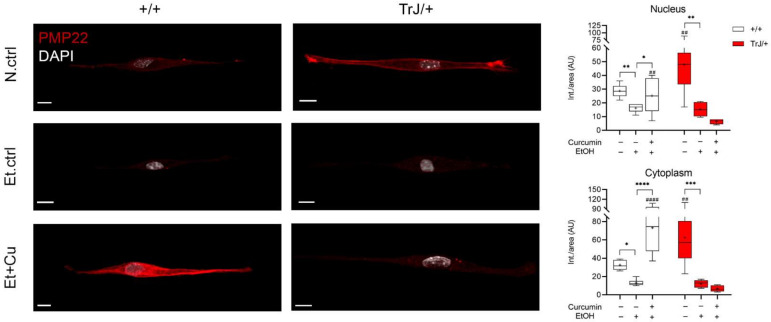
Modulation of PMP22 expression in +/+ and TrJ/+ SCs enriched cultures. Confocal microscopy images of +/+ and TrJ/+ SCs N.Ctrl, Et.ctrl (0.43 mM EtOH) and Et+Cu (0.25 µM curcumin + 0.43 mM EtOH) were quantified at the nuclear and cytoplasmic level. Et.ctrl vs. N.ctrl, had a nuclear decrease in PMP22 expression in +/+ SC, while in TrJ/+ SC it was observed in both compartments. On the other hand, Et+Cu treatment vs. Et.ctrl, showed an increase compared to both cell compartments, only for the +/+ genotype. In turn, the comparison +/+ vs. TrJ/+ SCs of the same condition revealed differences in both compartments, in N.ctrl and Et+Cu treatment. Within each genotype: * *p* < 0.033; ** *p* < 0.002; *** *p* < 0.0002; **** *p* < 0.0001. Same treatment between +/+ and TrJ/+: ## *p* < 0.002; #### *p* < 0.0001. The mean is shown as “+”. Scale = 10 µm for all panels. n = 100 cells on average per condition for each genotype.

**Figure 5 biomolecules-12-00515-f005:**
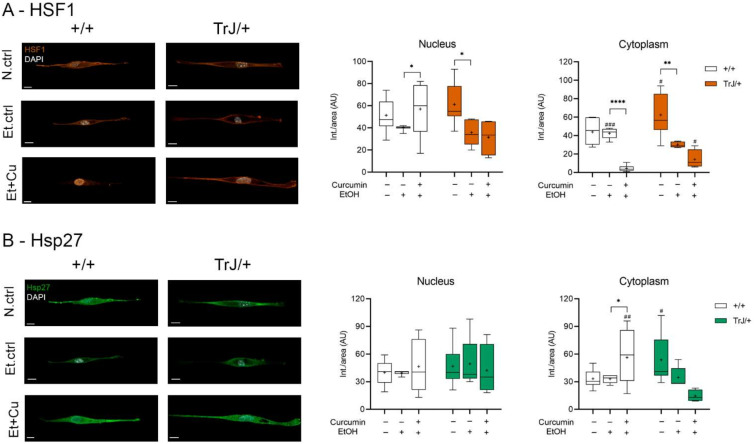
Modulation of HSF1 and Hsp27 expression levels in +/+ and TrJ/+ SCs enriched cultures. Images obtained by confocal microscopy of +/+ and TrJ/+ SC N.ctrl, E.ctrl, and Et+Cu were quantified in the nucleus and cytoplasm. (**A**) HSF1 expression. Et.ctrl vs. N.ctrl had a decrease only in TrJ/+ SC in both compartments. Et+Cu treatment vs. Et.ctrl showed a nuclear increase and a cytoplasmic decrease only in +/+ SCs. In turn, the comparison +/+ vs. TrJ/+ SCs of the same condition revealed only cytoplasmic differences in Et.ctrl and Et+Cu; (**B**) Hsp27 expression. No differences were observed between Et.ctrl vs. N.ctrl, in both genotypes or both compartments. A comparison +/+ vs. TrJ/+ SCs of the same condition revealed cytoplasmic differences in N.ctrl and Et.ctrl. Within each genotype: * *p* < 0.033, ** *p* < 0.002 and **** *p* < 0.0001. Same treatment between +/+ and TrJ/+: # *p* < 0.033, ## *p* < 0.002 and ### *p* < 0.0001. The mean is shown as “+”. Scale = 10 µm for all panels. *n* = 100 cells in average per condition for each genotype.

**Figure 6 biomolecules-12-00515-f006:**
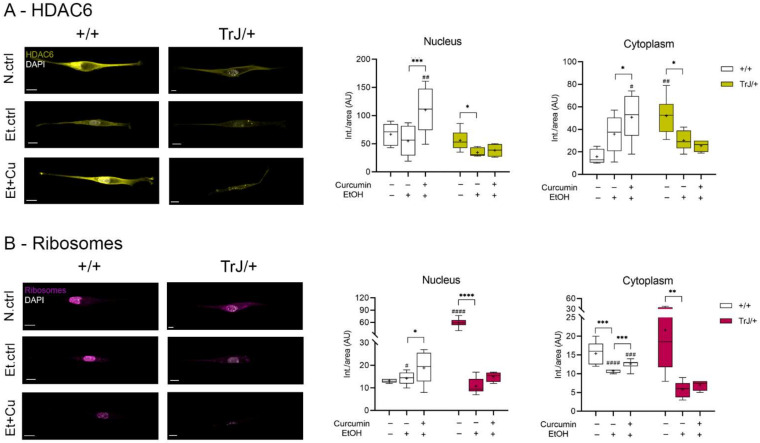
Modulation of HDAC6 and ribosome expression in +/+ and TrJ/+ SC enriched cultures. Images obtained by confocal microscopy of +/+ and TrJ/+ SC in N.ctrl, E.ctrl, and Et+Cu were quantified in the nucleus and cytoplasm. (**A**) HDAC6 expression. Et.ctrl vs. N.ctrl showed a decrease only in TrJ/+ SCs in both compartments. Et+Cu vs. Et.ctrl showed an increase in +/+ SCs in both compartments. Conversely, the same condition between +/+ and TrJ/+ SCs revealed nuclear differences only with Et+Cu. At the cytoplasmic level, differences were observed with the N.ctrl and Et+Cu; (**B**) Ribosome’s expression. Et.ctrl vs. N.ctrl, showed a nuclear decrease only in TrJ/+, while the cytoplasmic decrease occurred in both genotypes. Et+Cu vs. Et.ctrl showed an increased expression in +/+ SCs in both compartments. +/+ vs. TrJ/+ SCs revealed nuclear differences with the N.ctrl and the Et.ctrl, and cytoplasmic differences with Et.ctrl and Et+Cu treatment. Within each genotype: * *p* < 0.033; ** *p* < 0.002; *** *p* < 0.0002; **** *p* < 0.0001. Same treatment, +/+ vs. TrJ/+; # *p* < 0.033; ## *p* < 0.002; ### *p* < 0.0002; #### *p* < 0.0001. The mean is shown as “+”. Scale = 10 µm for all panels. *n* = 100 cells in average per condition for each genotype.

**Table 1 biomolecules-12-00515-t001:** Morphometric parameters of SC and FB. Evaluation of the area, major and minor length of Schwann cells (SC), and fibroblasts (FB) of +/+ and TrJ/+ cultures. Measurements expressed in Mean ± SEM.

Cell Type	Area (µm^2^)	Major Length (µm)	Minor Length (µm)
FB	180,554 ± 27,043	615 ± 61	275 ± 41
SC	6062 ± 699	456 ± 51	30 ± 3

## Data Availability

Data available on request due to restrictions e.g., privacy or ethical. The data presented in this study are available on request from the corresponding author.

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
