# Peer review of "Curcumin and Ethanol Effects in Trembler-J Schwann Cell Culture"

_biomolecules, 2022, doi:10.3390/biom12040515_

Round 1
Reviewer 1 Report
This is an interesting paper, meant to describe the mechanisms of action of Curcumin on Schwann cells in culture, obtained from WT and TrJ mice. However, I have some remarks to formulate.
Remark 1
The solvent used in the study (100% Ethanol) to dissolve Curcumin seems to have blocked any therapeutic effect of Curcumin. If I fully agree with the authors with their statement that the solvent of molecules may interfere with the effects of the molecules themselves, I think that 100% Ethanol was probably not the best choice to get Curcumin dissolved as it may contain toxic molecules. As you may know, in order to get 100% Ethanol, you have to perform an azeotropic distillation, i.e. you have to add benzene to the mix of water/ethanol. Traces of this molecule may be found in the distillate. Thus, I would have used 96% Ethanol instead. Of course, I am not sure this is the cause of the negative results on TrJ cells but one may conceive this toxic molecule may have impacted on cells with an already impaired metabolism (TrJ).
A huge amount of good work has been done in this paper, so I am not asking to redo all of the experiments but I think it would be wise to pick a few of them to check if 100% Ethanol is not the cause of these results. And I would also do some experiments in parallel with curcumin dissolved in DMSO (mentioned in the paper but not presented).
Remark 2
Although it is interesting to work on cultured Schwann cells, which are primarily involved in Type I CMT forms, I think cultured cells do not behave similarly when not doing their physiological function, i.e. myelinating axons. In a future work, I would try to induce a myelinating differentiation by adding co-cultured neurons.
Remark 3
The natural half-life of Curcumin is usually quite short, sometimes as short as 30 minutes. So, I wonder if dosages of Curcumin in the medium have been made, and of curcuminoids. The dosages are not easy to perform I have to admit and most of the time require mass spectrometry. But since we do not see any effect of Curcumin on these cells, normal ones or TrJ ones (Figure 3, L308), I think this would be wise to check.
Remark 4
As stated by the authors, ethanol may have toxic effects on its own on mitochondria in general and even more on mitochondria of already impaired cells (TrJ). I agree with the authors that ethanol may have stimulated deshydrogenase (L530) rather increased viability, although this would have not bothered me in my personal life. This study will be important to keep in mind for preparing the formulation of Curcumin products.
Author Response
Reviewer 1: point-by-point responses:
Comment 1- The solvent used in the study (100% Ethanol) to dissolve Curcumin seems to have blocked any therapeutic effect of Curcumin. If I fully agree with the authors with their statement that the solvent of molecules may interfere with the effects of the molecules themselves, I think that 100% Ethanol was probably not the best choice to get Curcumin dissolved as it may contain toxic molecules. As you may know, in order to get 100% Ethanol, you have to perform an azeotropic distillation, i.e. you have to add benzene to the mix of water/ethanol. Traces of this molecule may be found in the distillate. Thus, I would have used 96% Ethanol instead. Of course, I am not sure this is the cause of the negative results on TrJ cells but one may conceive this toxic molecule may have impacted on cells with an already impaired metabolism (TrJ).
A huge amount of good work has been done in this paper, so I am not asking to redo all of the experiments but I think it would be wise to pick a few of them to check if 100% Ethanol is not the cause of these results. And I would also do some experiments in parallel with curcumin dissolved in DMSO (mentioned in the paper but not presented).
Response 1.- We thank the reviewer for this important remark. The effect of ethanol is seen in both genotypes. However, the wt genotype shows a metabolic recovery in the presence of curcumin dissolved in alcohol, which the TrJ genotype is not able to develop. As commercial preparations of absolute ethanol, may contain traces amounts of other components such as benzene or methanol, we cannot rule out that the effects observed are related to the trace compounds rather than the ethanol itself. However, the rather high ratio ethanol:trace elements (>1:1000) make this scenario highly unlikely. We have revised the chemical details of the ethanol used in these experiments. The alcohol used in our experiments is from the company DORWIL (brand anhydrous absolute alcohol, catalog no.: D010-00-03). The chemical characteristics of this alcohol can be accessed in the following link: http://www.dorwil.com.ar/pdf/CATALOGO%20DORWIL.pdf in page 9. That benzene detection test was performed according to the British Pharmacopoeia Ed. 1973., being the limit of benzene in the alcohol >2 ppm.
On the other hand, the curcumin’s manufacturer only mentions the final concentration in the ethanol (10 mg/ml, https://www.sigmaaldrich.com/UY/es/product/sigma/c1386), but does not provide information about the percentage. However, Beevers et al. (2006 PMID: 16550606) show the effects of curcumin dissolved in 100% ethanol in rhabdomyosarcoma. Regarding the Curcumin-DMSO experiments, we included preliminary results with DMSO concentrations in primary culture of fibroblasts obtained from wt sciatic nerve as supplementary materials. These results made us to reject the possibility of using DMSO as a vehicle of curcumin, because all the concentrations analyzed showed significant differences in both viability and proliferation assays when compared to the untreated control.
Comment 2.- Although it is interesting to work on cultured Schwann cells, which are primarily involved in Type I CMT forms, I think cultured cells do not behave similarly when not doing their physiological function, i.e. myelinating axons. In a future work, I would try to induce a myelinating differentiation by adding co-cultured neurons.
Response 2.- We share the reasoning of the reviewer and we totally agree on the fact that the used system does not entirely represent the nerve physiological context. However, it seemed to us that before moving to study the curcumin effects in a more complex model, as are the myelinating neuron/Schwann cell co-cultures, it was essential to investigate Schwann cells in a more simplified context. We agree with the reviewer that the model proposed in our work does not entirely resembles the physiological scenario in peripheral nerves, except for the fact that after nerve damage, the Schwann cells respond in an integral way, giving rise to various processes, including proliferation, to repair and re-establish the nerve homeostasis. After we determine a vehicle of curcumin with no per-se effects, we are planning in follow-up experiments to test the curcumin effect on dorsal root ganglion-Schwann cell co-cultures under proliferation and myelinating conditions.
Comment 3.- The natural half-life of Curcumin is usually quite short, sometimes as short as 30 minutes. So, I wonder if dosages of Curcumin in the medium have been made, and of curcuminoids. The dosages are not easy to perform I have to admit and most of the time require mass spectrometry. But since we do not see any effect of Curcumin on these cells, normal ones or TrJ ones (Figure 3, L308), I think this would be wise to check.
Response 3.- Together with its low availability, the short half-life of curcumin certainly represents a major challenge. We fully agree with the reviewer's remark and understand that these problems will continue to be present regardless of the model on which we work. However, we consider futile to do this dosage of curcumin with ethanol or DMSO, because we know that these vehicles have toxic effects per-se and the curcumin half-life analysis has no value in these conditions. We are currently initiating studies of curcumin dosage by HPLC and mass spectrometry, but these experiments are carried out with a different vehicle (data not shown).
Comment 4.- As stated by the authors, ethanol may have toxic effects on its own on mitochondria in general and even more on mitochondria of already impaired cells (TrJ). I agree with the authors that ethanol may have stimulated deshydrogenase (L530) rather increased viability, although this would have not bothered me in my personal life. This study will be important to keep in mind for preparing the formulation of Curcumin products.
Response 4.- We are grateful for the reviewer's comments and suggestions, as they have been valuable input to improve the work and to propose new perspectives.
Reviewer 2 Report
In this manuscript the authors evaluated the neurotherapeutic action of low dose curcumin treatment on the PMP22, HSF1 and Hsp27 expression and autophagy/mTOR pathways in Trembler-J and wild-type genotypes. They found that in wild-type Schwann cells, curcumin treatment stimulated chaperone and macroautophagy pathway while showed a mild effect on ribophagy. Moreover, despite the neuroprotective effects of curcumin for the treatment of neurological diseases, they found that curcumin effects in Trembler-J Schwann cells could be impaired due to the irreversible impact of ethanol used as a common curcumin vehicle necessary for administration.
Major Concern:
- Although the manuscript focuses on PMP22, the function of this protein is never described neither in the introduction nor in the discussion.
- Lines 516-517: Authors stated that DMSO showed the same ethanol effects but did not shown any data. However, this is a key point of this paper since DMSO is used as inert vehicle in spontaneously immortalized schwann cell line RSC 96 (PMID: 28622711) and in many other cell lines. It is further important since ethanol modulated many of the protein analyzed by the authors and, importantly cell proliferation. At least the data regarding DMSO and cell proliferation should be reported to validate this paper.
Minor Concerns:
- Line 100: it should be added that curcumin also showed anti-inflammatory effects at 10 µM decresing STAT3 activation (PMID: 31781039)
- In figures with mmunofluorescence, negative controls for PMP22, HSF1,HDAC6, Hsp27 must be shown
Author Response
Reviewer 2: point-by-point responses:
Major Concern:
Comment 1.- Although the manuscript focuses on PMP22, the function of this protein is never described neither in the introduction nor in the discussion.
Response 1.- In accordance with the reviewer's suggestion, we have incorporated more information on the function of the PMP22 protein in the introduction (lines: 53 to 70).
Comment 2.- Lines 516-517: Authors stated that DMSO showed the same ethanol effects but did not show any data. However, this is a key point of this paper since DMSO is used as inert vehicle in spontaneously immortalized schwann cell line RSC 96 (PMID: 28622711) and in many other cell lines. It is further important since ethanol modulates many of the proteins analyzed by the authors and, importantly cell proliferation. At least the data regarding DMSO and cell proliferation should be reported to validate this paper.
Response 2.- We than the reviewer for pointing this out. We have included preliminary results with DMSO on primary culture of endoneural sciatic nerve fibroblasts obtained from wild-type mice in the supplementary material (figure S2).
Minor Concerns:
Comment 3.- Line 100: it should be added that curcumin also showed anti-inflammatory effects at 10 µM decreasing STAT3 activation (PMID: 31781039).
Response 3.- The citation suggested by the reviewer was added with the number 64 and the information is now included in the text (Lines 115 and 116). Of note, the suggested paper does not mention the catalog number of the curcumin used (ignoring its degree of purity), nor does it state the vehicle used to dissolve it.
Comment 4.- In figures with immunofluorescence, negative controls for PMP22, HSF1, HDAC6, Hsp27 must be shown
Response 4.- Immunolabeling controls have been added as supplementary material, in figure S1 and mentioned in section 2.6 (Lines 252 and 253) in the main text.
Round 2
Reviewer 2 Report
The manuscript is significantly improved and can be accepted for publication in the present form.
Comment- The manuscript is significantly improved and can be accepted for publication in the present form.
Response- Thanks.